# The Importance of the Bursa of Fabricius, B Cells and T Cells for the Pathogenesis of Marek’s Disease: A Review

**DOI:** 10.3390/v14092015

**Published:** 2022-09-12

**Authors:** Karel A. Schat

**Affiliations:** Department of Microbiology and Immunology, College of Veterinary Medicine, Cornell University, Ithaca, NY 14853, USA; kas24@cornell.edu; Tel.: +1-607-351-3252

**Keywords:** Marek’s disease, bursa of Fabricius, pathogenesis, bursectomy, cyclophosphamide

## Abstract

The importance of the bursa of Fabricius (BF) for the pathogenesis of Marek’s disease (MD) has been studied since the late 1960’s. In this review, the results of these studies are analyzed in the context of the developing knowledge of the immune system of chickens and the pathogenesis of MD from 1968 to 2022. Based on the available techniques to interfere with the development of the BF, three distinct periods are identified and discussed. During the initial period between 1968 and 1977, the use of neonatal bursectomy, chemical methods and irradiation were the main tools to interfere with the B lymphocyte development. The application of these techniques resulted in contradictory results from no effects to an increase or decrease in MD incidence. Starting in the late 1970’s, the use of bursectomy in 18-day-old embryos led to the development of the “Cornell model” for the pathogenesis of MD, in which the infection of B lymphocytes is an important first step in MD virus (MDV) replication causing the activation of thymus-derived lymphocytes (T cells). Following this model, these activated T cells, but not resting T cells, are susceptible to MDV infection and subsequent transformation. Finally, B-cell knockout chickens lacking the J gene segment of the IgY heavy chain gene were used to further define the role of the BF in the pathogenesis of MD.

## 1. Introduction

Marek’s disease (MD) is a lymphoproliferative disease of chickens caused by Gallid alphaherpesvirus 2 belonging to the genus *Mardivirus* and commonly referred to as Marek’s disease herpesvirus serotype 1 (MDV or MDV-1) [1]. Since the isolation of MDV by Churchill and Biggs [2] in 1967, the importance of the bursa of Fabricius (BF) for the pathogenesis of MD became a topic of intense research and controversial results, especially during the period of 1967 to 1975. In this review, I will discuss the state-of-the-art of the knowledge of bursa- and thymus-derived lineages of lymphocytes until 1975 and the different methods used to interfere with the development of bursa-derived lymphocytes. This is followed by a description of the knowledge of the pathogenesis of MD during the same period. The importance of the BF for the pathogenesis of MD will discussed from 1967 until the current state-of-the art. It is important to realize that many of the results were obtained using methods that were state-of-the-art at the time of publication and that these methods evolved over the 65-year period covered in this review.

## 2. The State-of-the-Art of Avian B and T Cells from 1956 to 1970

In this section, I will briefly describe the knowledge on the importance of the BF and the thymus up to 1966, which is essential to understand the early studies on the role of the BF in the pathogenesis of MD. The terminology used in the next section reflects the understanding of immunology during this period. In some instances, the current terminology is included.

The bursa of Fabricius in chickens was described for the first time by Hieronymus Fabricius of Aquapendente (1537–1619), a professor of surgery at the University of Padua, Italy, as a hollow structure connected by a duct to the proctodeal region of the cloaca (reviewed in [3]). The function of the BF remained a mystery until 1956, when Glick et al. [4] accidentally learned that the BF is involved in antibody production. Glick et al. needed high titered chicken antiserum against the O antigen of *Salmonella typhimurium*. They inoculated chickens from which the BF was surgically removed [bursectomized (Bx)] and intact chickens, left over from another experiment, with heat-inactivated broth cultures by the intravenous route. To their surprise, some of the Bx chickens died after the inoculation while 3 Bx chickens survived but did not produce antibodies in contrast with the intact birds. This serendipitous finding prompted a large experiment confirming the finding that Bx chickens did not produce antibodies. These results led to the concept of the division of lymphocytes into a lineage of antibody-producing cells (B cells) and “other” lymphocytes associated with the thymus (T cells). Like in mice from which the thymus was surgically removed [thymectomized (Tx)], neonatal thymectomy of chickens resulted in delayed homograft rejections but did not interfere with the normal development of antibodies against human γ-globulin [5]. Warner and Szenberg [5] concluded that a dissociation of immunological responsiveness exists in chickens with antibodies produced by BF-derived B cells and homograft rejection by T cells. The same authors also used hormonal bursectomy (HBx) [6] to study the function of the bursa. Inoculation of testosterone propionate in the allantoic cavity of 12-day-old embryos caused the absence of the BF in a number of chickens. Some of the so-called “bursaless” chickens also showed thymus atrophy. In contrast with intact controls, both types of bursaless chickens were unable to produce antibodies after injection with bovine serum albumin (BSA). The average time to reject skin homografts was similar in control and bursaless chickens but skin homografts were not rejected in the bursaless chickens with thymus atrophy. These experiments provided further evidence for their hypothesis that there is a functional difference between lymphocytes derived from the BF and the thymus.

Max Cooper and colleagues [7,8] examined the different functions of the BF and thymus in more detail using a mixture of treatments consisting of Tx, neonatal Bx (NBx) and sub-lethal X-irradiation (X) or combinations such as TxX and NBxX. Histological examination of the spleen indicated that the clusters of small lymphocytes were absent in the spleens of the TxX group, while germinal centers were absent in NBx chickens. The current understanding of these structures, often referred to as red and white pulp, respectively, is reviewed in [9]. The TxX and TxNBxX groups had also a significant lower number of mostly small circulating peripheral white blood lymphocytes (PBL) than the other groups at 7 weeks of age suggesting that most of these cells were thymus-derived. TxX resulted in a prolonged presence (>27 days) of skin and wattle homografts in some but not all birds while these grafts were rejected within 10 days in control and NBxX chickens. TxX treatment also resulted in a significant reduction of the graft-versus-host (GVH) response using the Simonsen assay. In this assay, peripheral blood from 3-week-old chickens representing the different treatment groups are injected intravenously in 14-day-old embryos. At embryonation day (ED) 19, spleens were harvested, and the degree of splenic enlargement reflected the GVH response. Inoculation of 40-day-old, NBxX and X chickens with BSA or *Brucella abortus* antigen showed that the NBxX group was unable to produce antibodies confirming earlier publications. TxX chickens had a significant decrease in antibody titers, which could not be explained at the time but is now known to be caused by the absence of T helper cells. So-called gamma M and gamma G globulins (IgM and IgY, respectively) were absent after hatching in NBxX chickens while IgM, but not IgY, was present in NBx chickens comparable to intact controls.

## 3. Methods to Interfere with Normal Bursal Development and Antibody Production

A short description of the different bursectomy techniques in MD research is important. Understanding the pitfalls of these techniques may explain some of the contradictory results published on the influence of the BF on the pathogenesis of MD.

Chemical ablation of the BF can be achieved by injection of chicken embryos at ED 5 with 0.63 mg of 19-nortestosterone in corn oil [10] or at ED 12 or 13 with 2 to 2.5 mg of testosterone propionate in the allantoic cavity [6,11]. This technique results in a loss of B cells and thus antibody production but up to 40% of the birds can still develop antibodies [12]. In addition, thymus atrophy has been reported as a consequence of HBx [6] interfering with separating the functions of BF and thymus.

Cyclophosphamide (CY), an immunosuppressive alkylating agent with known lymphocytotoxicity, has frequently been used to interfere with B cell development. Lerman and Weidanz [13] showed that injection of chicks with 4 mg CY at 1, 2 and 3 days post hatch caused significant reduction in the amounts of circulating IgM and IgG. Unfortunately, CY is rather toxic causing some mortality. Most of the surviving chickens were unable to mount antibody responses after immunizations with 3 antigens at 6 and 10 weeks of age. Linna et al. [14] reported a rapid depletion of thymocytes especially in the thymic cortex using 2 mg CY/chick for the first 4 days post hatching. This was a short-term effect and the thymus returned to normal morphology after 15 days of age. GVH responses and allograft rejection were not impaired at one month of age indicating that there was no long-term negative effect of CY treatment on cellular immune functions. On the other hand, antibody responses were permanently suppressed.

Surgical methods to eliminate B cells and antibody responses include neonatal bursectomy as described by Peterson et al. [15]. NBx is often used in combination with sublethal X-irradiation, because NBx alone does not always eliminate the development of germinal centers in the spleen and the production of antigen-specific antibodies. Injection of ^3^H-thymidine into the bursal parenchyma of newly hatched chickens showed that B cells were migrating at one day of age to the thymus and spleen [16]. A subsequent study in which 18-day-old embryos received an intrabursal injection with ^3^H-thymidine showed migration of lymphoid cells to the spleen and thymus within 24 to 48 h post labeling [17]. Clearly, B cells migrate to the spleen and thymus prior to hatching which explains the ability to mount a humoral immune response after antigenic stimulation.

Earlier, Cooper et al. [18] had shown that embryonal bursectomy (EBx) at ED 17 greatly increased the number of birds without germinal centers and usually eliminates primary and secondary immune responses. We used the technique for in ovo bursectomy as described in [19] with some modifications allowing surgery on 20 embryos/hour with a hatchability of 50 to 60%. Inoculation of our EBx chicks with BSA and MDV showed the absence of humoral antibody responses when the surgery was complete without leaving bursal remnants [20,21,22].

The development of transgenic chickens (reviewed in [23]) made it possible to generate knock-out chickens. Schusser et al. [24] used homologous recombination in primordial germ cells to remove the joining (J) gene segment of the IgY heavy chain. Homozygous chickens for the J gene segment knockout had normal BFs, but development of mature B cells and migration of B cells from the bursa was blocked. As a consequence, these birds did not develop germinal centers and were not able to produce antibodies after immunization. This development opened new avenues to study the importance of B cells for the pathogenesis of MD and other diseases.

## 4. The Pathogenesis of Marek’s Disease: The Early Years until 1975

As early as 1929, visceral lymphomatous tumors were described in several birds with “fowl paralysis” [25]. Histopathologic examination of these tumors revealed closely packed round cells with the appearance of lymphoid cells [26,27]. As such this disease was part of a group of diseases often referred to as the avian leukosis complex which included visceral lymphomas caused by filterable leukosis viruses, while tumors associated with fowl paralysis could not be reproduced with filtrates [28]. In 1967 and 1968, Churchill and Biggs [2] and Solomon et al. [29], respectively, isolated a cell-associated herpesvirus from intact tumor cells or PBL. MD could be reproduced with these isolates and MDV was reisolated from the experimentally infected chickens. The isolation of MDV allowed controlled experiments to study the pathogenesis of infection. Direct and indirect immunofluorescent assays (IFA) became the advanced tools to study the distribution of MDV in the different organs of infected chickens. In this review, I will focus on those papers examining lesions and or distribution of viral antigens in the BF, spleen and thymus. Prior to the isolation of MDV, Purchase and Biggs [30] noted atrophy of some of the follicles in the BF as early as 7 days post infection (dpi) with blood from birds previously infected with 5 different isolates. These lesions progressed and at 21 dpi many follicles showed atrophy of the bursal medulla and cortex. Unfortunately, there is no information on the presence or absence of infectious bursal disease virus (IBDV) in these isolates, which were all obtained from field cases. Jakowski et al. [31] also propagated field isolates, JM [32] and Conn-A isolates of MDV in chickens, using blood from these birds to infect experimental chickens. They noted a significant decrease in bursal size at 12 dpi. The small bursas showed necrosis in the follicles and in many instances complete absence of the follicles. Although interesting, these results may not reflect actual MDV-induced lesions because there is no information on the absence or presence of IBDV and/or chicken infectious anemia virus (CAV) in the inocula. The same group used a related MDV isolate, Conn-B, which caused hematopoetic destruction and a significant reduction in packed cell volume [33]. This virus was later shown to be contaminated with CAV [34].

Spencer and Calnek [35,36] developed indirect and direct IFA’s allowing the detection of MDV antigens in tissue sections of experimentally infected chickens. These birds were either infected by inoculation with JM-induced tumor cells, JM propagated in cell culture of by contact with JM-infected donors. Importantly, viral antigens were detected in the BF. At the same time, Calnek and Hitchner [37] and Calnek et al. [38] examined a large number of tissues for viral antigens obtained from chickens infected with tumor suspensions or by contact exposure to previously infected birds. Tissue samples were collected between 14 and 42 dpi or when birds were paralyzed. The BF was positive in many of these samples while tumors samples were frequently negative. Von Bülow and Payne [39] examined bursa tissues from 67 chickens between 12 and 56 dpi with cell culture propagated HPRS-16 or VC II MDV. In total 54% of the samples were positive for MDV antigen using a direct IFA. The positive samples also showed lymphocyte necrosis in the follicles while the interstitium was normal. Because IBDV can also cause necrosis of the follicles, samples were examined with a FA conjugate for IBDV and all samples were negative for that pathogen. Thus far, all studies were done with samples at 12 dpi or at later time points. Adldinger and Calnek [40] exposed chickens by inhalation to pulverized feathers from JM-infected donors at 7 or 28 days of age and examined tissues collected between 12 h and 22 dpi for the presence of viral antigen using a direct IFA and an agar gel precipitation (AGP) assay. The latter test was used to detect soluble antigens in tissue extracts. Between 3 and 5 dpi the AGP assay was positive for extracts from the BF, spleen and thymus when infected at 7 or 28 days of age. IFA tests for BF, spleen and thymus did not become positive in some birds until 5 dpi and at 6 dpi 3/4 birds were positive in the 3 lymphoid organs when exposed at 28 days of age. Viral antigens were intermittently detected in the BF (e.g., at 10 dpi). Viral antigens were frequently detected from 16 to 41 dpi in the BF but sporadically in the thymus and spleen. Purchase [41] infected one-day-old or 3-week-old chickens with JM-positive blood or JM-infected duck embryo fibroblasts and examined the BF, thymus and other organs for MD antigens using an IFA and an AGP test. In addition, he examined tissues for histopathologic lesions. Starting at 5 dpi, the BF was positive for viral antigen in the IFA and at the same time bursa samples showed some evidence of necrosis. Although the thymus samples were also positive for viral antigens in some birds, there is no information on the dpi when these samples were positive.

In conclusion, these combined results indicated that the BF as well as the thymus are important organs for the pathogenesis of MD. However, the data did not allow to draw specific conclusions on the actual role of the BF and thymus in the development of MD. The possibility to use any of the techniques to eliminate the BF and to a lesser degree the thymus became an important tool to further study the pathogenesis of MD and to identify the cell types which are important for viral replication and tumor formation.

## 5. The Use of Chemical or Neonatal Bursectomy Techniques to Investigate the Role of the Bursa of Fabricius in Marek’s Disease

Foster et al. [42] used in ovo injection of 4 mg testosterone propionate at ED 12 to examine the importance of the BF for MD. Chicks were challenged at one day of age with plasma from young, contact-infected game-fowl chickens with clinical MD. The authors claimed that HBx reduced the incidence of MD. However, 37 of the 52 HBx chickens died between 6 to 8 weeks of age due to infection with β-hemolytic streptococci while the observation period for MD was 20 weeks. Thus only 15 chickens were at risk for the development of MD with 4 chickens developing MD for an actual incidence of 27%, which is not significantly different from the 39% MD incidence in the control group (data re-analysis by Schat using the Chi-square test with Yates correction). Thus, the correct conclusion is that HBx did not change the outcome of MDV infection. In addition, the use of plasma, which is supposed to be free of cells, is not a good source for MDV challenge, because MDV is a strictly cell-associated virus. Carte et al. [43] compared MD incidence between intact controls, NBx chickens and chickens after intramuscular injection of 10 mg testosterone propionate at 7 days of age. All birds were challenged at 7 days of age with a 20% MD tumor suspension and the experiment was terminated at 12 weeks of age. The authors claimed that NBx caused the highest incidence of MD at 20.3 % versus 14.8 and 11.1% for the HBx and intact challenged groups, respectively. However, the abstract did not provide a statistical analysis and the non-challenged control group had also a 4.1% incidence of MD.

Neonatal bursectomy without (NBx) or with additional sub-lethal X-irradiation (NBxX)was used by three groups of researchers. Morris et al. [44] found that NBx increased the MD incidence and reduced the mean time to death. Unfortunately, the results may be confounded by using whole blood from chickens previously infected with MD, which does not exclude the possibility of co-infection with immunosuppressive viruses such as CAV and other immunosuppressive viruses (reviewed in [45]). CAV has been linked to suppression of T helper cells as well as cytotoxic T cells (CTL) and the CTL are very important for the immunity to MD [46]. In 1970, Payne and Rennie [47] used NBx or NBxX chickens infected with blood of MDV- infected donors (experiment 1) or cell culture-propagated virus (experiment 2). In both experiments, most of the NBx and NBxX chickens were unable to mount an antibody response to *Salmonella typhimurium*-O antigen, Brucella abortus antigen, sheep red blood cells (SRBC) and BSA. There was no difference in overall MD, although the figures suggest a slight delay in the development of gross MD mortality in the NBxX versus the sham-operated control group. It is of interest to note that Payne and Biggs mentioned in an earlier publication [48] unpublished data from Payne, Witter and Burmester suggesting that NBxX ameliorated the incidence of MD, which was addressed as not confirmed in the 1970 paper. In the same year, Fernando and Calnek [49] also used NBx or NBxX chickens but exposed the birds by contact exposure or intra-tracheal inoculation with cell-free virus. Most NBx and NBxX chickens were unable to produce antibodies to Salmonella pullorum antigen and BSA indicating that the treatments were successful. The results were very similar to the results reported by Payne and Rennie [47]. Both groups concluded that the BF was not essential for the development of MD thus eliminating the hypothesis that B cells could be target cells for transformation as had previously been suggested by the Calnek group [36,37].

Since the reports that injections of CY shortly after hatch interferes with the development of the BF and the ability to produce antibody responses in chickens [13,14], it became a tool to study the role of the BF in MD tumor incidence and pathogenesis. Purchase and Sharma [50] used line 7 and 15 × 7 chickens, which are highly susceptible to MD, to examine the effects of CY on MDV-induced mortality, gross lesions and total incidence of MD including histological lesions. Combined data for exposure to the JM and GA strains of MDV at 6 or 21 days of age showed a significant decrease in the CY-treated groups for mortality and gross lesions. However, there was no significant difference between control and CY-treated groups when histological lesions were included in the analysis. The CY treatments were effective based on the absence of antibody responses to SBRC. Because CY affects the thymus for the first 10–14 days after the treatment, it was not clear, if the rather high dose of 16 to 20 mg CY/bird may have accounted, at least in part, for the reduction in mortality and gross lesions. Sharma and Witter [51] challenged control and CY-treated line 15 × 7 chickens with JM clone 19 MDV at 8 or 9 weeks of age to determine if the BF was important for age-related resistance to MD. Although in both experiments the MD incidence was lower in the CY-treated group than in the controls, the differences did not reach the *p* = 0.05 level of significance. However, the results were considered significant at *p* < 0.05 when the two experiments were combined. Controls included the GVH response to test for thymus-derived functions which was not affected by the CY treatment when tested at the time of challenge with MDV. The effects on bursal functions were examined by the morphology of the BF, absence of germinal centers in the cecal tonsils and the inability to produce antibodies to MDV. The morphology of the BF showed the typical lesions caused by CY such as the absence of lymphoid cells in the follicles. Germinal centers in the cecal tonsils were absent in 45/51 CY-treated chickens and MDV antibodies were absent in 44 of these chickens, which is in contrast with the control infected chickens. Clearly, antibodies did not play a role in the age resistance to MD. Kermani-Arab and collaborators [52,53] used relatively resistant and susceptible chickens, which were treated with CY during the first 4 days of age with doses from 5 to 16 mg CY/bird. Mitogen stimulation of spleen cells to Concanavalin A indicated a normal T cell function in the CY group, but B cell functions were ameliorated in the birds receiving 16 mg of CY with an absence of germinal centers in the cecal tonsils and spleens, the inability to produce antibodies to SRBC and the presence of bursal lesions. MDV challenge with 1500 plaque-forming units (PFU) was performed at 3 weeks of age when maternal antibodies to MDV were absent. Virus isolations at 7 and 14 dpi were markedly reduced in the birds with the higher dose of CY (16 mg/bird) but were no longer significantly different from the control infected group starting at 21 dpi. CY treatment with 16 mg/bird reduced the MD incidence including microscopic lesions in MD susceptible birds and delayed the development of MD lesions. The following suggestions were made to explain the results in the susceptible birds: (1) the absence of antibodies to MDV would allow cell-mediated, tumor-destructive immune responses to predominate and (2) CY treatment may have reduced the number of susceptible cells for virus replication resulting in fewer cells becoming infected for viral transformation resulting in a delay of the onset of tumor development [53]. Interestingly, the authors actually dismissed this idea based on unpublished data from their own laboratory that chickens treated with a high dose of CY and observed for 8 weeks post infection still had a low incidence of MD lesions. It is difficult to analyze the importance of these unpublished data without more information. A PubMed search for the authors of the unpublished data (Lu, Kermani-Arab and Moll) did not yield additional relevant information.

Two groups also reported information on the role of the BF in vaccine-induced immunity to MDV challenge. Purchase and Sharma [50] treated chickens with 20 mg CY/bird and vaccinated these birds with herpesvirus of turkeys (HVT) at 6 days of age followed by challenge at 3 weeks of age. CY treatment abolished protection in two experiments. The authors concluded that the absence of protection was most likely due to the obliteration of the bursa-dependent lymphoid cells although they did not completely rule out that some residual thymus effector cells caused some protection. In hindsight it is more likely that the early effects of CY treatment on the thymus prevented the development of a strong vaccine-induced CTL response. In support of the latter explanation Else [54] found that NBxX chickens were fully protected against challenge with HPRS-16 after being vaccinated with HPRS-16att indeed suggesting an important role for cell-mediated immune responses rather than antibody-mediated protection.

## 6. The Use of Embryonal Bursectomy to Study the Pathogenesis of MD

The approaches described in the previous section did not yield conclusive results on the role of the BF in the pathogenesis of MD. This was in part due to the fact that B cells migrate to the periphery starting at ED 18 [17] and the transient depressive effect of CY on the thymus. Complete embryonal bursectomy (EBx) at ED 17 eliminates the migration of B cells from the BF to the periphery and thymus functions are not affected. Sharma [55] used EBx to determine if the genetically MD resistant line 6 chickens would become susceptible in the absence of B cells. Eleven of the 16 EBx chickens did not have any bursal remnants. These 11 chickens lacked germinal centers in the spleen, did not produce antibodies to SRBC, and IgM and IgG were not detectable in serum in contrast to the five incomplete EBx chickens. MDV challenge with JM at one week of age resulted in viremia in 4/11 completely EBx chickens but none of the 11 birds or sham-operated line 6 chickens developed MD. Susceptible line 7 chickens inoculated with the same inoculum developed 90% MD indicating that the challenge dose was capable of causing MD. The results clearly showed that the genetic resistance of line 6 chickens is not related to the ability to produce antibodies.

For my PhD thesis research [56] I started to modify the EBx technique described in [19] allowing large numbers of embryos to be bursectomized with 50 to 60% hatchability. The application of EBx became an important tool in our departmental research group to investigate not only the pathogenesis of MD but also for studies on CAV [57] and avian rotavirus [58] in chickens and reticuloendotheliosis virus in ducks [59]. Our results using EBx for the MD research are detailed below.

Calnek et al. [20] used EBx, CY and Tx to determine if the low virulent strains CU-1, CU-2 and S11 have a low inherent transformation potential or that the low pathogenicity is caused by generating a superior host immune response. All EBx chickens were free of bursal remnants. The CY-treated chickens had marked atrophy of the bursa and the follicles were mostly devoid of lymphocytes. Both groups of chickens were unable to develop antibodies to MDV and BSA. Thymectomy was often incomplete with remnants mostly present in the thoracic inlet and there was no difference in skin allograft rejection between the control and Tx groups. EBx and CY treatment did not increase the pathogenicity of CU-1 and CU-2, nor did it decrease the pathogenicity of these two viruses. The MD incidence in the controls was already very low and a decrease would not be significant anyway. In the case of challenge with S-11, CY and to a lesser degree EBx decreased the incidence of MD compared to controls but the differences were not significant. In contrast, Tx increased the MD incidence, which included the development of lymphomas.

The importance of the BF and thymus for the rejection of a MD transplantable lymphoma was examined in genetically resistant N-line chickens using GA/Tr-1 [21], which was subsequently renamed as MDCT-CU8 [60]. Intact, EBx, CY, Tx, Tx/CY and NBx/CY treated chickens received the tumor cells in the breast muscle at 7 days of age and the experiments were terminated at 10 weeks of age. EBx and CY treatments resulted in the absence of antibody development while Tx/CY but not Tx caused a significant decrease in mitogen stimulation of PBL and some prolongation of skin graft rejection. In the intact birds, palpable tumors in the breast muscle had a mean time to regression (MTR) of 16 days and at 14 days post injection only in 12 % of the chickens showed tumors regression. At 10 weeks of age, the total tumor incidence was 53%, which included 6% with pectoral muscle tumors at the site of inoculation. Tx and Tx/CY resulted in MTR of 18 and 26 days with 0% regression at 14 days post injection. Total incidence of MD was very similar to the control group with 58 and 69%, respectively. The results for EBx and CY treatments were very interesting with MTR of 13 and 17 days, respectively, and 67% regression for the EBx group at 14 days. The total incidence of MD at 10 weeks was 6% in the EBx and CY groups. Infection with MDV at one day of age followed by injection of the transplantable tumor resulted in a significant prolongation of the MTR showing the impact of early immunosuppression by MDV. Two explanations were offered for the enhanced rejection and significant reduction in overall MD incidence especially in the EBx group. The first one was that these treatments eliminated “blocking” antibodies resulting in an enhanced T cell response causing a more efficient rejection of the transplantable tumor cells. However, efforts by the authors to demonstrate blocking of the rejection by injecting EBx chickens with MDV or “anti-tumor” serum failed to delay rejection of the transplantable tumor cells. A second explanation was that bursectomy eliminated putative “suppressor T-cells”, which following Droege may have been derived from the BF [61].

Since the 1970’s the pathogenesis of MD in susceptible chickens has been described with four distinctive phases (reviewed in [62]). The early cytolytic phase is characterized by a productive-restrictive infection in the lymphoid organs as early as 2 to 3 dpi with extensive virus replication in the BF, thymus and spleen. This is followed by a latent infection with little or no evidence of viral replication. The third and fourth phase consist of a late cytolytic infection resulting in permanent immunosuppression and finally tumor development. Thus far, most if not all publications on the effects of bursectomy on MD examined the outcome of infection in terms of mortality and MD lesion incidence. In some instances, viremia results were included but not for the early cytolytic phase. Yet, the findings with MDCT-CU8 suggested that the early cytolytic phase could be important for the pathogenesis. Schat et al. [22] examined in a number of experiments the importance of the BF for the early and late cytolytic infection. EBx and intact P-2 chickens were infected at 2 weeks of age and samples for virus isolation and IFA were collected between 3–11 dpi. Birds with bursal remnants were excluded from the results. In three experiments, the spleens and thymuses were positive for viral antigens in the intact but negative in the EBx birds between 3–6 dpi. Viral antigens were not detected in thymuses and spleens from both groups at 7–11 dpi suggesting that latency had been established. Viral isolations from PBL, spleen and thymus cells from EBx chickens were mostly negative between 3–6 dpi in contrast with the samples from intact controls. The relative weights of the spleens were significantly enlarged in the intact infected chickens compared to uninfected intact and EBx controls between 4–11 dpi. The spleens from infected EBx chickens were comparable to the non-infected control groups. MDV-induced splenic enlargement between 3–11 dpi had previously reported by us [63]. In contrast with serotype 1 MDV, the early pathogenesis of the non-oncogenic, serotype 2 SB-1 strain [64] was not altered by EBx. In both groups, virus was isolated at similar levels from spleens and PBL, viral antigen expression was limited to a few scattered cells in the spleen and thymus, and spleens were significantly enlarged compared to intact controls as previously published [63]. EBx also affected the long-term outcome of infection with a slightly prolonged mean time to death and a significant decrease in total MD incidence after challenge with JM. Several explanations were offered to explain the results. Two of the explanations, absence of blocking antibodies and BF-derived suppressor T cells, were discussed before. A third possibility was a putative suppressor B cell, which had been described in mice [65]. However, these cells had not been described for chickens at that time or since then to my knowledge. The final explanation that oncogenic MDV strains need an early replication in B-cells, which then carry infection to transformable T cells, was rejected based on the knowledge of the early pathogenesis in 1980. However, soon afterwards it was learned that this explanation was actually correct.

## 7. The Development of the Cornell Model for the Pathogenesis of MD

It is important to realize that the work presented in this section was performed when the first monoclonal antibodies for chicken lymphocyte markers were just being developed. Fluorescence activated cell sorting (FACS) analysis and separation of chicken cell populations had not really started in the period between 1978 and 1982 and that MDV strains expressing fluorescent markers were not available until much later. Moreover, cytokines that might be used to cultivate chicken lymphocytes in vitro were not yet discovered. Relevant cytokines to cultivate B cells did not become available until approximately 40 years later (see Section 8).

The finding that the early pathogenesis of MD was altered in EBx chickens prompted Shek, a PhD student in the department, to characterize the type of cells infected with MDV in the BF and thymus during the early cytolytic infection and latency [66,67]. The methodology to separate B cells from T cells and macrophages started with the isopyknic separation of JM-infected spleen and thymus cells in Percoll gradients. During the lytic infection, virus infection based on virus isolation was mostly associated with the low-density fractions while the virus-positive cells during the early latency were mostly associated with the mid-density fractions, a shift that was statistically significant at *p* < 0.01. These results suggested that the cells involved in the early cytolytic infection were different from the latently infected cells. When these experiments were in progress, we obtained the first monoclonal antibodies against the chicken μ-chain of IgM (MAC IgM) [68] and chicken Ia-like antigens (MAC Ia) [69] from Max Cooper’s laboratory through a very productive collaboration with Dr. Chen-lo Chen. Ia -like antigen is actually MHC class II antigen [70] which was shown in 1984 to be present on all B cells and activated T cells [69]. Ficoll-Paque-treated cells were fractionated by incubation to carbonyl iron, adherence to nylon wool or adherence to IgG-coated SRBC. Carbonyl iron-treated, nylon wool non-adherent cells and Fc-negative cells were depleted of IgM+ and Ia+ cells compared to nylon-wool adherent and Fc-receptor positive cells. Infectivity at 3–5 dpi was associated with the latter group of cells (*p* < 0.01). Most latently infected cells did not adhere to nylon wool and were Fc-receptor negative. These results suggested that the early cytolytic infection is associated with B cells and the latently infected cells with non-B-cells. Using MAC Ia and complement it was then shown that most of the MDV-positive cells were Ia-positive at 3–5 and 19 dpi. Using a B-cell rosetting technique [71] with some modifications we were able to demonstrate that the infectivity at 3 and 5 days was associated with the B cell enriched, rosetted fractions but with the non-rosetted fractions at later time points. Although these results suggested that B cells are the major target cells for the early cytolytic infection, it was not clear why the thymus had so many positive cells during the early infection.

The techniques described above were also used to examine the possibility to infect lymphocytes in vitro [72]. Spleen lymphocytes or specific subsets of spleen lymphocytes were exposed to MDV using three different methods. Freshly prepared spleen lymphocytes or subpopulations were co-cultivated for 48 h with 48-h old spleen cell cultures from MDV-infected birds or with MDV-infected chick kidney cells (CKC). In addition, spleen cell were exposed to cell-free JM. Freshly prepared spleen cells were added at 48 to 72 h intervals at a ratio of 3 to 1. Cell viability dropped off rapidly and cell viability was about 10% after 72 h in culture. Successful virus infection was determined by virus isolation in CKC or by examining lymphocytes for the presence of viral antigen by IFA with the latter assay being about thousand-fold more sensitive than virus isolation. We were able to passage virus from spleen cells to spleen cells for at least 43 passages when the experiments were terminated. Successful virus transfer was independent of the genetic resistance to MD tumor development showing that lymphocytes from the resistant N-2 line with MHC B^21^B^21^ were not more resistant to virus infection than lymphocytes from susceptible P-2 (MHC B^19^B^19^) line. Transfer of virus was similar when syngeneic or allogeneic spleen cells were added to the cultures. Using the techniques described in [67] we showed that the infected cells were B cell based on the presence of Fc receptors, surface IgM and surface Ia antigen. This observation was further substantiated when the addition of spleen cells from EBx donors reduced the level of infection significantly. Addition of thymocytes from intact donors resulted in a significant lower number of virus-infected cells compared to spleen cells. This is not surprising because Shek et al. [67] had shown that virus-positive cells in the thymus are actually B cells, which constitutes a minor population in the thymus. Based on these two papers [67,72] combined with the study on the pathogenesis in EBx chickens [22], it can be concluded that B cells are the major target for the early cytolytic infection. It was suggested that this approach could be used for in vitro transformation of lymphocytes, which was indeed achieved in 1991 when we reported that 3/122 attempts resulted in proliferating lymphoblastoid cells. These cells resembled conventional MD lymphoblastoid cell lines expressing the pan-T cell marker CD3 and Ia [73].

The development of MD lymphoblastoid cell lines was first reported in 1973 when Akiyama et al. [74,75] established two cell lines with MSB-1 becoming an important tool for MD research. Shortly afterwards, Powell et al. [76] established two cell lines which were characterized as T cells. We established six more cell lines, which were T cell lines [77] followed by several more cell lines from lymphomas from MHC-defined chickens infected with different MDV strains varying in virulence [78]. These cell lines (n = 31), MSB-1 and RP-1 [79] were characterized by us using some of the approaches described by Shek et al. [67]. All MD cell lines were negative for surface IgM using MAC IgM and Fc receptors but were positive for Ia antigen with the exception of RP-1 and the ALV-positive cell line LSCC-CU10 [80]. When more monoclonal antibodies against chicken lymphocyte markers became available, we revisited the characterization of transformed cells. We examined 44 MD lymphoblastoid cell lines established from lymphomas and 56 cell lines developed from local lesions induced by intramuscular or wing-web inoculation with allogenic MDV-infected CKC [81]. All cell lines were positive for Ia antigen, CD3 and/or T cell receptors (TCR). Cell lines developed from tumors were CD4^+^ except one, which was CD4^−^CD8^−^. TCR2 (TCRαβ1) was expressed on 84% of these cell lines while 16% expressed TCR3 (TCRαβ2). The cell lines established from the local lesion model represented a mixture of CD4^−^CD8^+^ (45%), CD4^+^CD8^−^ (21%) and CD4^−^CD8^−^ (34%) genotypes [82]. These results confirmed that MD cell lines are indeed activated T cells. More recently, tumor cell lines and tumor cells have been classified as T regulatory (Treg) cells based on molecular characterization [83].

The findings that the early cytolytic infection is mostly occurring in B cells based on in vivo and in vitro experiments, but that tumor cells are activated T cells led to the question how to explain the apparent switch from B cells to T cells.

To address this question Calnek et al. [84] infected P-2, N-2 or UCD-003 chickens with MDV at 3 or 4 weeks of age and spleens were harvested between 1 and 5 dpi for the early cytolytic infection and at 22 dpi to obtain latently infected cells. Spleen lymphocytes were virus positive as early as 1 to 2 days post infection based on the detection of viral antigens, but expression of viral antigen at day 1 and 2 was only detectable after cultivating the cells for 48 h. Spleen cells were examined for viral antigens and cell surface markers by dual immunofluorescence assays using MAC IgM and MAC-T, which detects CD3, for surface markers and an anti-MDV conjugate for the detection of viral antigens. Between 3–4 dpi 83 to 92 % of the MDV-positive cells were B cells and a small minority of 2 to 3% were T cells while the rest of the MDV-positive cells could not be typed perhaps because the cell surface markers were already lost. In contrast to the cytolytic infection, latently infected cells were separated using rosetting assays with MAC IgM, MAC T and MAC Ia to enrich or deplete B cells, T cells and Ia-positive cells. Lymphocytes were then cultured for 48 h to reactivate viral replication and thus antigen expression. The large majority of the latent cells were identified as Ia-positive (90.8%) and non-B cells (97.6% positive). T cell enriched cells had 70% viral antigen positive cells. These results were further confirmed using dual fluorescence assays with 84 to 95% of the viral antigen positive cells expressing CD3, but there was a small number of antigen positive cells expressing surface IgM. The overall results of these experiments showed that the early cytolytic infection were mostly B cells and the latently infected cells mostly activated T cells with a small percentage of B cells. In vitro infection studies were conducted to determine if T cells could maintain an in vitro infection cycle using the approach described before. Donor spleen cells came from intact or EBx birds. Unfortunately, some of the EBx birds had a very low percent IgM-positive cells, but when donor cells came from a bird lacking all IgM cells, it was shown that lytic infection cycle could be maintained in activated T cells based on the expression of CD3 and Ia antigen [85]. Finally, we showed that resting T cells are mostly refractory to infection in contrast to activated T cells [86].

In conclusion, we showed that (1) the early cytolytic infection occurs mostly in Ia-positive B cells and in a small number of activated T cells, (2) that latent infection is mostly present in Ia-positive T cells with a very minor component in B cells, (3) tumor cell lines are T cells expressing Ia antigen and (4) that resting T cells seem to be refractory to infection in contrast with activated T cells expressing Ia. These conclusions led to the “Cornell model” of the pathogenesis as reviewed in detail in [62,87]. Recently, Baaten et al. [88] showed that early replication of MDV after exposure by the respiratory route occurred in pulmonary B cells further demonstrating the importance of the B cells for the pathogenesis of MD.

The importance of the spleen for the pathogenesis was examined by using splenectomy (SX) [89]. Spleens were surgically removed from 7- to 9-day old chickens. Intact and SX P-line chickens were infected by the intra-tracheal route using cell-free JM-10 or GA-5 strains. The efficacy of the splenectomy procedure was evaluated by visual inspection at necropsy and if needed tissues were collected surrounding the usual location of the spleen. Birds with residual splenic tissue were excluded from the results. The early pathogenesis was examined by virus isolation of PBL and detection of viral antigens in the BF and thymus by IFA. SX birds were negative at 4 and 6 dpi for virus isolation and viral antigen while intact controls were positive at 4 dpi. Starting at 7 dpi, the SX chickens were positive for virus isolation and viral antigens. The development of tumors was the same after infection with GA-5 (100% in both groups). Interestingly, JM-10 induced significant fewer tumors in SX (33%) than in intact chickens (78%) (*p* < 0.001). However, almost all SX chickens without lymphomas had A-type lesions in the nerves so that the total MD incidence was similar in both groups. These results suggest that cell-free virus may reach the spleen first perhaps after replication in the pulmonary B cells [88] or after transport by macrophages to the spleen. The difference in lesion development after JM-10 infection between the two groups could not be explained.

## 8. The Use of Cytokines to Study the Pathogenesis of Marek’s Disease

The use of CD40 ligand (CD40L) [90] to cultivate B cells combined with T cells stimulated with monoclonal antibodies to TCR2 (anti-αVβ1) [91] allowed Schermuly et al. [92] to develop an efficient in vitro infection model for MDV. CD40L stimulated B cells were successfully infected by co-cultivation with MDV-infected chicken embryo fibroblast cultures. Stimulated T cells were also infected in vitro using the same procedure used for B cell infection. Interestingly, two populations of infected T cells were present: CD4^−^TCR^−^ T cells were lytically infected, while the CD4^+^CD8^−^ TCR^+^ T cells were latently infected. Infected B cells were able to transfer MDV to stimulated primary thymus-derived T cells and long-term cultures of MDV positive T cells were established without the need for stimulation. These cells had the characteristics of cell lines developed from MDV tumors. These results confirm and extend the studies done by Calnek and associates in the early 1980’s. Berthault et al. [93] noted that in vivo infection caused the high degree of apoptosis in the thymus and bursa, causing atrophy as has been shown by many others. However, the unexpected finding was that the apoptosis in the BF occurred mostly in the MDV-negative cells in contrast to the apoptosis in the thymus, which occurred in the MDV-infected cells. Unfortunately, the authors did not characterize the apoptotic cells in the thymus as B or T cells. This is an important question because as shown in the previous section, many MDV-positive cells in the thymus are B cells. Perhaps the microenvironment in the thymus is not conducive to protect MDV-infected B cells from apoptosis. The observation that MDV-infected B cells survived in the BF while uninfected B cells became apoptotic and cell proliferation declined raised several important questions. Trapp-Fragnet et al. [94] used the in vitro infection model to confirm that MDV-infection increased the viability of B cells by infecting B cells as described in [92] and after 24 h MDV-positive and negative cells were sorted and kept in culture without CD40L. MDV-infected B cells survived significantly longer than the control cells suggesting that these cells are either proliferating or prolonging their survival. Microarray analysis of MDV-positive versus negative cells showed that the cells were not proliferating but were surviving longer developing a snescence-like phenotype. The authors conclude that the senescence prolongs the time for MDV to complete their replication cycle enhancing the dissemination of virus to T cells recruited to the BF.

## 9. The Use of Transgenic Chickens in Marek’s Disease

The transgenic chicken line JH^−/−^ lacking the ability for B cells to migrate from the BF developed by Schusser and coworkers [24] was recently used to further examine the importance of the BF and B cells for the pathogenesis of MD [95]. Wild-type, JH^−/−^ and JH^+/−^ chickens were infected at one-day of age with the very virulent RB-1B MDV strain [96] and evaluated for virus replication and tumor formation. Infection was established in the BF at 4 dpi and at much lower levels in the thymus (*p* < 0.05) and spleen, but these differences were no longer detectable at 7–10 dpi. Virus presence in the BF of the JH^−/−^ was comparable with that in the JH^+/−^ chickens. It is certainly possible that the viral replication in the BF attracted T cells, as has been described for IBDV [97], followed by subsequent transfer by T cells to other locations. Although the tumor incidence of MD was lower in the JH^−/−^ than in the wild-type chickens and the total MD incidence was higher in the JH^−/−^ than in the wild-type chickens, these differences were not statistically significant. These results validate and complement the results in EBx chickens [22] and the subsequent studies by Calnek et al. as detailed in Section 7. In wild-type chickens, the infection of B cells remains an important part for the pathogenesis.

## 10. Conclusions

The B cells play an important role during the early pathogenesis of infection in intact chickens resulting in the development of innate and acquired immune responses. The latter will result in the development of activated T cells, which can be infected and some of the infected T cells may develop into tumor cells. The finding that MDV expresses v-IL8, a functional ortholog of the chicken CXCL13 chemokine [98], is important because the expression of vIL-8 allows the infected cells to attract uninfected B and activated T cells to transfer the virus to new cells [99]. One of the unanswered questions that still needs to be addressed is how the MDV CD4^+^ T cells are pushed into latency and what determines which of these cells transform into tumor cells.

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
