# Peer review of "The Importance of the Bursa of Fabricius, B Cells and T Cells for the Pathogenesis of Marek’s Disease: A Review"

_viruses, 2022, doi:10.3390/v14092015_

Round 1

Reviewer 1 Report

The author has set out to review the role of the bursa of Fabricius and bursal-derived B-lymphocytes in the pathogenesis of Marek’s disease, dividing the review into sections based on the stepwise development (over the course of 65 years) of new techniques to study bursal function and thereby to study the role of the bursa in MD.  The author has discussed and reinterpreted results of early studies in the context of more recent data.  This is a very comprehensive and thorough review bringing together, for the first time, the history of research into the role of the bursa and B cells in MD, to give an updated summary of the current knowledge of MD pathogenesis.  I enjoyed reading the review, it refreshed my memory about past research, and I gained new information and insight.

The review is written very clearly, and is well-referenced and easy to read.  I have no major criticisms, just a list of suggestions for improvement, and some minor corrections.

(1)    The contents of the review are broader than the title suggests.  From the title, I was expecting it to focus predominantly on the bursal organ, its role as a target for MDV infection, and also as the source of B cells for the humoral immune response against MD.  The main focus of the first half of the review is on B cells, so I felt ‘B cells’ should also be included in the title.  There is also a lot of information on the thymus and T cells.  In the later part (development of the Cornell model) the review drifts further from the title as there is much more focus on T cells and transformation (although closely interlinked with the bursal/B cell research).  Therefore, I felt the title of the review could be amended to better reflect the fact that the review covers the bursa, thymus, spleen, B and T cells.

(2)    Lines 44-46: It would be useful to briefly explain the location and structure of the bursa (although I appreciate the author has referenced a review on this).

(3)    Lines 49 and 56: It would be useful to briefly define bursectomy and thymectomy here, in case any reader is not aware that it refers to destruction/removal of the bursa/thymus.

(4)    Conclusions: The conclusions don’t really match the title as there is no mention of the bursa in the conclusions (as in #1 above, modifying the title would address this).

(5)    It would be useful to make clear that the bursa/B cells have a role as both the primary target cells for MDV, but also as a defence against MD, and that pathogenesis is influenced by the balance in these two roles.

(6)    Although not essential, it would be nice to include some diagrams and images to break up the text.  For example: a diagram to summarise the history/timeline of methods researching the role of the bursa/B cells in MD; a diagram to summarise the current understanding on the role of the bursa/B cells in MD.

 Minor corrections:

(7)    In the title and line 8, bursa has an uppercase ‘B’.  Elsewhere it has a lowercase ‘b’ (correct).

(8)    In the first part of the manuscript, ‘Tx’ is used as the abbreviation for thymectomy.  Later, ‘TX’ is used; I’m not sure if this should be ‘Tx’ or whether it refers to thymectomy with X-irradiation.

(9)    Line 110: Spelling error – ‘morphology’.

(10)Line 116: Spelling error – ‘eliminate’.

(11)Line 118: Repetition of ‘3H-thymidine’.

(12)Line 175: Spelling error – ‘The’.

(13)Line 183: How were the chickens ‘exposed to pulverized feathers’? (Inhalation? Inoculation?)

(14)Line 215: Spelling error – ‘conclusion’.

(15)Line 221: Spelling mistake – ‘highest’.

(16)Lines 237-238: Should this say ‘gross MD tumours’ (rather than ‘gross MD mortality’?)

(17)Line 280: Spelling mistake – ‘plaque’.

(18)Line 290: Spelling mistake – ‘interestingly’.

(19)Line 294: Spelling mistake – ‘without’.

(20)Line 299: Spelling mistake – ‘vaccinated’.

(21)Line 299: I suggest writing HVT in full as I think this is the first time this abbreviation has been used in the manuscript.

(22)Line 299: Should this read ‘6 days of age’ (instead of ‘6 dpi’?)

(23)Line 401: Change ‘had not described’ to ‘had not been described’.

(24)Line 444: I suggest writing CKC in full as I think this is the first time this abbreviation has been used in the manuscript.

(25)Line 521: Change ‘latently’ to ‘latent’.

(26)Line 529: Was splenectomy performed in ovo or after hatch?

(27)Line 664: Spelling mistake - ‘infected’.

(28)Line 745: Change ‘Shek WRC, C.L.’ to ‘Shek WR, Chen CL’.

(29)I suggest adding the following to the abbreviations list: HVT, FACS, CKC.

Author Response

Please see the attchment

Reviewer 2 Report

Dr. Schat nicely reviews the importance if the bursa of Fabricius for MDV infections of chickens.

Here are some suggestions that should be addressed during preparation a next version of this review article.

Line 25f and elsewhere: The current ICTV nomenclature assigned the Mardiviruses into species rather than serotypes!

Please revise the sentence starting in line 117: “Injection of 3H-thymidine into the bursal parenchyma of newly hatched chickens with 3H-thymidine showed that B cells were migrating at one day of age to the thymus and spleen [16].”

Wouldn’t it be helpful to briefly describe the infectious route of the virus in the introduction so that the readers can understand MDV infection via “pulverized feathers” (line 183f)?

Line 217: …add “as MDV is strictly cell associated” or something similar.

Line 411f: These cytokines are now available and some studies made use of cultured bursal B cells that were infected with MDV. That could be discussed and some of the data presented (see INRA/LMU Munich papers below). I am listing examples of additional (recent) INRA papers below that could be discussed briefly because of interesting data on bursal B cells and MDV infections:

1.           Berthault C, Larcher T, Härtle S, Vautherot JF, Trapp-Fragnet L, Denesvre C. 2018. Atrophy of primary lymphoid organs induced by Marek's disease virus during early infection is associated with increased apoptosis, inhibition of cell proliferation and a severe B-lymphopenia. Vet Res 49:31.

2.           Trapp-Fragnet L, Schermuly J, Kohn M, Bertzbach LD, Pfaff F, Denesvre C, Kaufer BB, Hartle S. 2021. Marek's disease virus prolongs survival of primary chicken B-cells by inducing a senescence-like phenotype. PLoS Pathog 17:e1010006.

Line 555f: The JH-/- birds seemed to have slightly higher MD incidences. Please double-check and revise accordingly.

Line 558: As far as I’m aware, the authors (from Kaspers/Schusser/Kaufer labs) state that their data “provide important insights into MDV pathogenesis” and challenge “the crucial role of B cells” (because mature B cells indeed are dispensable/irrelevant for viral replication in vivo, as well as for MD and MDV-induced tumor formation) and not that their “findings changed the understanding of the pathogenesis of MD”. Please revise/correct.

Typos: immunosuppressive (line 102), morphology (line 110), irradiation and eliminate (line 116), consequence (line 136), immunofluorescent (line 151), The BF (line 175), 2x between (line 184), intermittently (line 190), conclusion (line 215), highest (line 221), possibility (line 229), immunosuppressive (line 230), plaque (line 280), predominate and (line 288), in a (line 290), Interestingly (line 290), without and authors (line 294), vaccinated (line 299), 2x the (line 301), embryonal (line 309).

Round 2

Reviewer 2 Report

Dr. Schat sufficiently addressed most raised issues and improved the manuscript.

One last thing that needs to be taken care of are changes that should be done to section 9 of this review:

Dr. Schat changed the wording a bit but I think that the statement in the second last sentence of section 9 (“The conclusion by the authors that their findings changed the dogma of the central role of the B cells in the pathogenesis is not supported by their results.”) should be toned down for the following reasons:

First, it has to be clarified that Dr. Schat compares reference 22 (Schat KA, Calnek BW, Fabricant J. Influence of the bursa of Fabricius on the pathogenesis of Marek's disease), a paper that shows MD data after removal of a whole organ, to data obtained from genetically engineered chickens that lack mature and peripheral B cells. In the PNAS paper, no differences were observed in MD, tumor incidence, and tumor dissemination using the vv+ RB-1B strain of MDV. That difference and the respective results should be reflected in section 9.

In reference 22, which Dr. Schat refers to as “similar results”, he states that

a)       The early lytic infection in the lymphoid organs normally associated with oncogenic MD virus infection in intact chickens was not seen in EBX chickens.

b)      …and a lower or delayed MD mortality when compared with intact chickens.

using oncogenic MDV.

Additional relevant Calnek-papers that Dr. Schat refers to in section 7 include references 67 (Shek et al., 1983), 72 (Calnek et al., 1982), 84 (Calnek et al., 1984a), and 85 (Calnek et al., 1984b) – all of which nicely show that B cells are major target cells for early MDV infection (and that some T cells support early lytic replication), while later T cells are latently infected.

For example, in line 521f it says that “it was shown that lytic infection cycle could be maintained in activated T cells based on the expression of CD3 and Ia antigen” which is what has also been observed in the JH-KO paper.

Dr. Schat ends a central paragraph of this section with “further demonstrating the importance of the B cells for the pathogenesis of MD” (see line 533).

The JH-KO paper also shows that T cells support early lytic replication but also shows that (i) the early lytic infection in/spread to the lymphoid organs can be observed in JH-KO chickens and that (ii) JH-KO chickens do not show lower or delayed MD mortality when compared with wt chickens (in contrast to reference 22, see above) as two exemplary additional features of the B-cell knockout.

Therefore, I suggest that the concluding sentences of section 9, starting in line 597, should be changed to “These results validate and complement the results in EBx chickens [22] and subsequent studies by Calnek et al. as detailed in section 7. In wild-type chickens, however, the infection of B cells remains an important part of the viral replication cycle.” – and hope that the author agrees with this.

Author Response

The suggested change has been made and is highlighted in green.
Hopefully the manuscript will now be accepted.